# Developing Guidelines for the Use of Passive Thermography on Cultural Heritage in Tropical Climates

**Manogna Kavuru** [1] and **Elisabetta Rosina** [2,*]

1   Center for Heritage Conservation, College of Architecture, Texas A&M University,
    College Station, TX 77840, USA; kamanog@tamu.edu
2   Department ABC, Politecnico di Milano, 20133 Milano, Italy
*   Correspondence: elisabetta.rosina@polimi.it

**Abstract:** Infrared thermography (IRT) has been a very successful tool for the diagnosis and monitoring of cultural heritage restoration projects. It has been used to identify anomalies, moisture issues, etc., in historic buildings. Although it is a promising tool, one of the limitations is that a method to deploy it onsite has not been standardized. This is due to the different variables that might affect thermal signatures captured by the thermal camera, when onsite. Especially since environmental conditions play a major role in thermography, the process must vary from region to region significantly. That said, efforts have been made over the years to establish some base standards for designated purposes of infrared thermography in the construction field. These standards and best practice methods, although comprehensive, do not effectively help with issues that are contextual to the location of the building, for instance, tropical climates, such as India. This paper aims to suggest guidelines for a passive approach of thermography, based on practical applications and procedures followed during the thermographic survey at the former British Residency in Hyderabad, India. Additionally, this paper explores the avenues through which region specific guidelines can be established.

**Keywords:** passive infrared thermography; guidelines; diagnosis; monitoring; restoration

## 1. Introduction

Infrared thermography (IRT) has been a very successful tool for the diagnosis and monitoring of cultural heritage. In restoration projects and the planning of maintenance, IRT has been increasingly utilized since the early 1970s for its application on historical buildings [1]. It has been deployed to identify structural elements underneath the finishing, moisture issues, and other anomalies in historic buildings [2–6]. It is also extensively used due to its nondestructive nature, which is based on images, and it can be used on a wide surface in a short time and from a distance.

Thermography for moisture monitoring of buildings is well-documented in existing literature [7–11], and it is a very useful technique, especially for preliminary scanning and for monitoring the surface in intervals of time. It does not require sampling and, therefore, can also be applied on precious surfaces. Although it is a promising tool, one of its limitations is that when it is deployed onsite, the results are affected by different variables. Uncontrolled conditions of air temperature, relative humidity, solar irradiation, etc., surrounding the target surface, affect thermal signatures captured by the thermal camera. Especially if the location of the building is in tropical climates, such as India, the climate could prevent reliable results because of the higher humidity levels [12].

The passive approach is defined as the application of the technique without an artificial/external heating (or cooling) of the surface and it is particularly suitable for the detection of humid surfaces at

certain ambient condition [13–15]. A passive approach is particularly convenient in the preliminary phase of the analysis, when the target of the investigation is any thermal anomaly that should be evaluated in a second step, comparing the information acquired by IRT, and the data resulting from the historical, visual analysis, the surveys, and any further exam on the site, and in the laboratory.

Although the climatic condition of a place could generally be considered suitable for the optimal application of the IRT, it is mandatory to record the air temperature and relative humidity, solar irradiation, wind speed, etc. [15], at the very moment of the recapture. The environmental data are strategic both for the recapture and for further analysis. In fact, the analysis has the aim of detecting a false alert due to anomalies of the temperature distribution that are not dependent on the object of the investigation, but on the boundary conditions and their changes affecting the measurements.

In the case of tropical climate, the regularly high RH, especially during the monsoon conditions, requires monitoring the environmental condition of the testing site. Collecting the historic climate and the local measurements in the days before the test (few days–one week, depending on the time window for testing) is very useful when considering, in advance, the possibility of success of the test, and to mitigate (if possible) the adverse condition. Moreover, in the case of building materials at balance with the environmental conditions, the surface temperatures are very similar. Therefore, any thermal anomaly that does not correspond to any visual anomaly has to be considered an alert signal in the first phase of the analysis.

Qualitative observation of the temperature pattern is the first filter for detecting the anomalies, due to the visualization at false colors of the surface temperature map. Nevertheless, the qualitative analysis is only the first step for obtaining the required information from the thermograms. In fact, after the comparison between the visual and the infrared images, further analysis of the anomalies must be accomplished. The interpretation of the thermograms require an interdisciplinary work: history, building technology and science, and evaluation of damages, are necessary for evaluating each thermal anomaly.

This study adds to pioneering work in establishing IRT as nondestructive testing used for diagnostics that could be applied to the preservation of built cultural heritage, without a big budget and with limited time. Especially since the advantages are clearly recognized, it is important that the procedure be exported to other contexts, which would benefit from such tools [6,16]. Hyderabad in India, being one such context, is discussed in this paper, to pave the way towards establishing region-specific guidelines that can serve as a basis to design thermal surveys in other parts of the country, where thermal imaging for the preservation of built heritage is still in its nascent stages.

*Background*

The British Residency in Hyderabad was commissioned in 1803 (by the then Nizam of Hyderabad) for it to be used as a residence for the British envoy to the court of the Nizam (first representing the East India Company and later the British government), (as shown in Figure 1). James A. Kirkpatrick was the British envoy responsible for the construction of the Residency [17]. While construction of the building, the interiors, and later refurbishments were funded by the Nizam, the building and its premises were under the control of the British, which is quite possibly the reason for the neoclassical architecture of the building. The building was completed by Lieutenant Samuel Russell of the Madras Engineers in 1806. The process of construction can be viewed as two separate processes, where the design of the building was largely based on British ideologies of beauty, while the construction and the building materials had to be local due to transportation issues. Post-Indian independence, most of those buildings have been re-purposed and have been adapted to the growing infrastructural needs of the country [18]. The British Residency is representative of all those buildings that were constructed in India during the period of the British in India. When the British left Hyderabad in 1949, the Residency was handed over to the Women's College of Osmania University (established in 1924). What is also significant is the location of the building along the River Musi in the heart of the bustling city of Hyderabad, which has caused the British Residency to be interwoven tightly into the historic fabric of

the city. Thus, the project of restoration at the Residency is prominent in the country, not only due to the importance of the site, but also due to the coming together of various national and international agencies. The Heritage Department of the Government of Telangana, Osmania University College for Women, and the World Monuments Fund, represent some of the highest standards of preservation practice in the country [19].

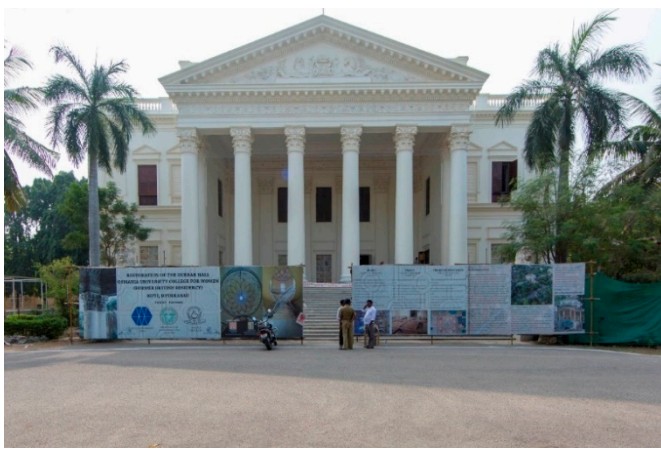

**Figure 1.** Northern facade of the British Residency in Hyderabad (image courtesy of the site conservation team, GN Heritage Matters).

Hyderabad Climate

Hyderabad is located in the southern plateau region of India. It is generally hotter than most parts of northern India. Based on the Köppen Classification System, Hyderabad falls under the tropical wet and dry climatic region of India. The climate varies significantly according to the seasons. According to the Indian Meteorological Department, Hyderabad is comprised of four seasons—winter (December to February), summer (March to May), monsoon (June to August), and post-monsoon (September to November). May is the hottest month, with temperatures reaching 40 °C, and December is the coldest month, with temperatures dropping to 15 °C. The thermal survey conducted at the former British Residency, Hyderabad, compares results from the monsoon to the summer reason (the wettest and the driest seasons) [20].

## 2. Materials

The iconic main central building of the Residency is being studied here; the other surrounding buildings on the sprawling 42-acre complex are not discussed in detail. The building is divided into three levels; the lower ground, upper ground, and the first floor levels [21,22]. The original construction of the Residency was of stone masonry at the lower ground level and brick masonry at the upper ground and the first floor. The lower ground area is a series of stone vaulted spaces covered in lime plaster, which were used as a treasury before the building was converted into the college. The superstructure has the durbar hall as a huge central volume, around which the other rooms are organized. The upper floors are made of brick masonry, and infilled arches indicate the change in the orientation of the building sometime during its history. The roof was mostly Madras roof (which is a combination of bricks and waterproofing layer made of traditional materials)—except for the central volume, or Durbar Hall, which is topped by a wooden vaulted roof, and the grand staircase, which has a concrete dome over it. Repair and additions to the building have been made from time-to-time, some not very compatible with the existing structure [19]. This has resulted in areas prone to moisture and structural damages, prompting the use of infrared thermography as an investigative tool to aid the conservation decision-making process. This is because of its proven history of being a successful diagnostic and monitoring tool in the field of heritage conservation, in many scenarios.

*Methods*

Thermal surveys (Thermal Camera FLIR T540 belonging to Texas A&M University was used for the survey. FLIR Tools was used to analyze the thermal images. Since there were no shiny surfaces in the vicinity causing reflections, reflective temperature is not considered for this analysis. Wind conditions are also not considered here as the images discussed are on the interior of the building) at the Residency were conducted by a team of researchers from Texas A&M University and Politecnico di Milano. The surveys were designed with the tropical wet and dry climatic conditions at Hyderabad kept in mind. Surveys were conducted in the month of August and February due to the vast difference of climatic conditions, being the wettest and the driest seasons in the region. While many guidelines and standards exist for the use of infrared thermography, not many elaborate on particular details, and this could be due to the ambiguity of the results presented by the instrument itself. The appearance of the same temperature pattern on a surface that can be due to different causes, depending on both the object of investigation and the boundary conditions of the surrounding environments, are possible reasons that lead to the ambiguity. Knowledge of the building, its history, and construction materials, in combination with a familiarity of the process of thermal imaging and image interpretation, is key to the success of a thermal survey. For instance, in this study, the help of reports from the team of architects working on the building and the expertise of the researchers about IRT was used to good effect. The team also tried to identify a few international standards that, while are not meant for the direct purpose of this survey, would still aid with some of the considerations to be made for the thermal survey. The American Society for Testing and Materials (ASTM) standard C1060 for the Thermographic Inspection of Insulation Installations in Envelope Cavities of Frame Buildings describes that the difference between interior and exterior temperatures of the building should be at least 10 °C for a 4 h period prior to the survey. The ASTM standard C1153 Standard Practice for Location of Wet Insulation in Roofing Systems Using Infrared Imaging suggests that no appreciable precipitation should occur 24 h prior to the thermal survey [23,24].

Emissivity was estimated to be about 0.95 using an insulation tape [14,25]. The type of day was noted to possibly correlate solar radiation to the results of the survey. With the average sunrise occurring at 6:00 a.m. in Hyderabad, over the course of the survey, the average sunrise time was noted to be 6:00 a.m., and care was taken to capture images at least 6 h after sunrise. This was done in order to allow the building to absorb solar radiation. In the interior, there is no incident sunlight, but this allows increased internal temperatures to be achieved on the inside, relative to the morning, decreasing humidity, and allows the most evaporative flux to take place from wet surfaces. While certain conditions of temperature gradient between the interior and exterior temperatures did not meet the requirements mentioned in the standards, the survey was continued for the purpose of understanding the error in results that suboptimal conditions would produce. The following are the conditions mapped for the two surveys (Table 1).

**Table 1.** Environmental parameters mapped for the thermal surveys.

| Conditions | Survey in August 2019 | Survey in February 2020 |
| --- | --- | --- |
| Ambient Temperature | 23 °C | 30 °C |
| Exterior Temperature | 25.5 °C | 32.7 °C |
| Relative Humidity | 96% | 50% |
| No. of Days after Rainfall | 0 | 14 |
| Typology of Day of Survey | Overcast day | Sunny day |

## 3. Results

Images from the two surveys produced very different images of the same areas. The images from August 2019, mostly, are indistinct in some areas and do not provide much clarity, while the images

from the survey in February 2020 produced much sharper images. The images of the left corner of the eastern arch, at the lower ground oval eastern room, and in the adjacent inside corner of the lower ground oval eastern room area are discussed (locations of which are marked in Figure 2), where direct effects of solar radiation are not considered, see Figures 3 and 4.

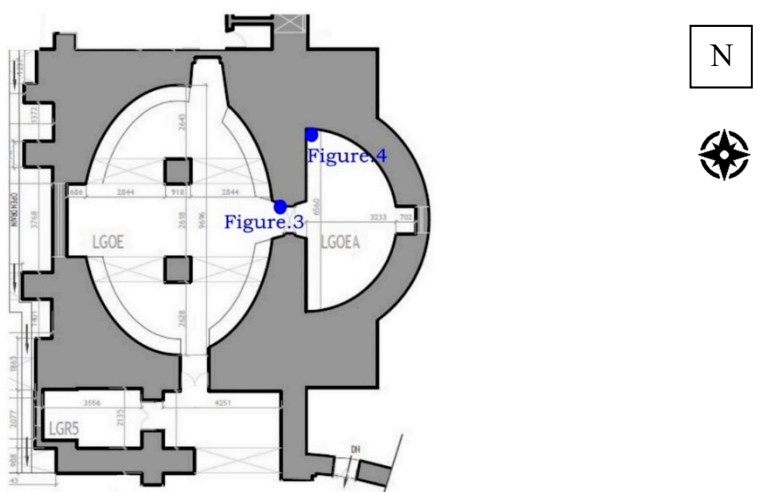

**Figure 2.** Part of the lower ground floor plan depicting the regions of Figures 3 and 4.

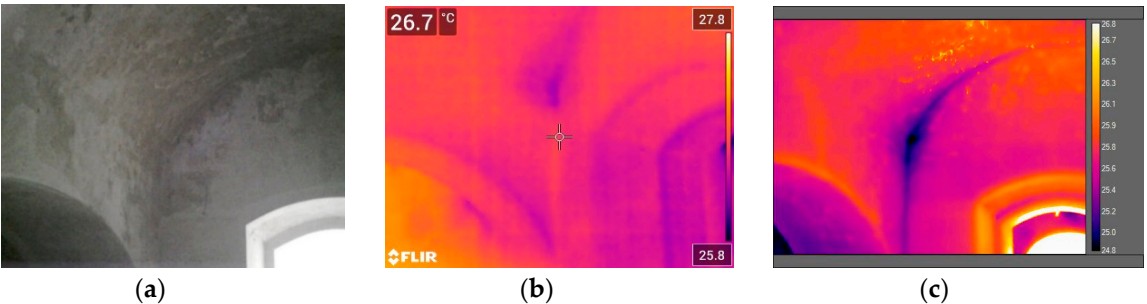

(**a**)　　　　　　　　　　　(**b**)　　　　　　　　　　　(**c**)

**Figure 3.** (Left to right) the set of images represent the left corner of the eastern arch in the lower ground oval eastern room. (**a**) Is the visual image of the area, (**b**) is the thermal image taken during the survey in August 2019, (**c**) is the thermal image taken during the survey in February 2020.

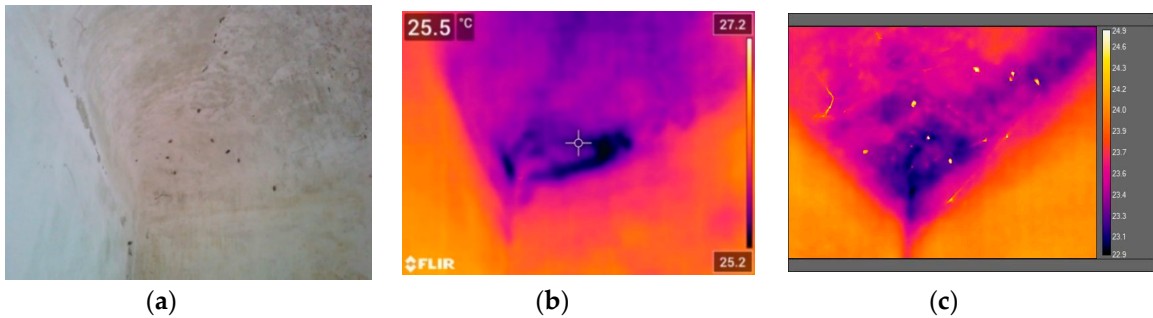

(**a**)　　　　　　　　　　　(**b**)　　　　　　　　　　　(**c**)

**Figure 4.** (Left to right) the set of images represent adjacent inside corner of the lower ground oval eastern room area. (**a**) Is the visual image of the area, (**b**) is the thermal image taken during the survey in August 2019, (**c**) is the thermal image taken during the survey in February 2020.

The two areas discussed here were subjected to a leaking pipe upstairs on the upper ground floor level, where a bathroom was added to the corner of the upper ground level oval room at some point when the building functioned as the College for Women. The images taken in August 2019 and February 2020 depict the resulting moisture from the aforementioned leak that was arrested just before

the thermal survey. Yet, it shows that the effects of the infiltration are still very much visible. It is the contrast with which the infiltration is visible that is the cause for discussion here. The images show clear distinction in the level of detail of the anomaly observed and smallest dissimilarity on the target surface.

One of the largest inferences made on this site versus prior studies was the large influence of Relative Humidity and the external temperature of the building on the imaging. This site being located in a tropical wet and dry climactic zone has high humidity when compared with temperate zones. Good IR images depend on the ability to acquire high contrasts in temperature between materials. This temperature contrast, or delta T, produces well-defined images that permit researchers to identify clear distinctions in images particularly at the vault junctions above.

The function of relative humidity in thermal imaging has been well-documented; high humidity reduces the surface evaporation of moisture from building materials. Water intrusion in porous materials change the thermal properties of the material itself. Therefore, the difference of temperatures between the wet and dry areas is minimized when the fluid flux is lowered due to high relative humidity [5,12,15,26]. Consequently, this reduction in fluid flux lowers the contrast available in thermal images and degrades their quality. Moreover, the evaporation flux occurring on the surface of porous materials cools the surface, up to 2 °C–3 °C. This change of the temperature, where the evaporation flux occurs, is clearly detectable by IRT. However, in this case, the surface is directly heated especially where the heating is not even (the surface temperature ranging more than 10 °C), the effects of cooling due to the evaporative flux are not detectable.

There are several standards developed to aid in thermal surveys and related imaging in broad and general terms. This study in India, particularly in Hyderabad, shows that there are some variables that can also impact the imaging process, and need to be considered for making a holistic approach to passive thermal imaging, especially in tropical climates. Based on this, we propose a modest rubric by which we can judge the efficacy of employing IRT at a site in different months in Hyderabad, see Figure 5. This rubric leans on the above-mentioned standards, as well as from our experience [23,24]. Moreover, year round temperature and relative humidity data for Hyderabad come from the Meteorological Department of India.

Based on ASTM Standard C1060, we can see that the ideal imaging temperatures result in a delta T between internal and external surface temperatures of 10 °C, which is the difference in temperatures between the internal surface to image, and the surface temperature of the exterior of the structure. Temperature gradients lower than this result in degraded imaging conditions due to the formation of misrepresentations in the images. These errors lead to inconclusive imaging and leave room for errors in image interpretation. However, in the case of the former British Residency, the thickness of the building does not allow for regular heating of the inside of the building directly, due to solar radiation, it is rather due to the rise of the exterior temperature that causes the increase in the interior temperature due to the ventilation in the building. The building itself takes much longer to heat or cool when relying on direct solar radiation. Hence, as the temperature rise inside the building occurs due to the increase in external temperature, this prompts increased evaporation, which in turn is the reason for visibility even at smaller temperature gradients. We understand that the exterior temperature is one of the most important parameters for a thermal survey in such climates.

The conclusion drawn from this survey, as well as prior experience, showed the researchers that relative humidity has similar impacts on imaging quality. There is an acceptable range for relative humidity as well. This led to the development of the above quadrant graph and rubric. Optimal conditions for thermal imaging can be defined as those with exterior temperatures higher than 20 °C and relative humidity of 50% for tropical climates. The least ideal conditions are those with very high humidity and very high temperatures, degrading the imaging environment completely. Non-ideal conditions are those with a higher temperature than standard or relative humidity greater than 50%.

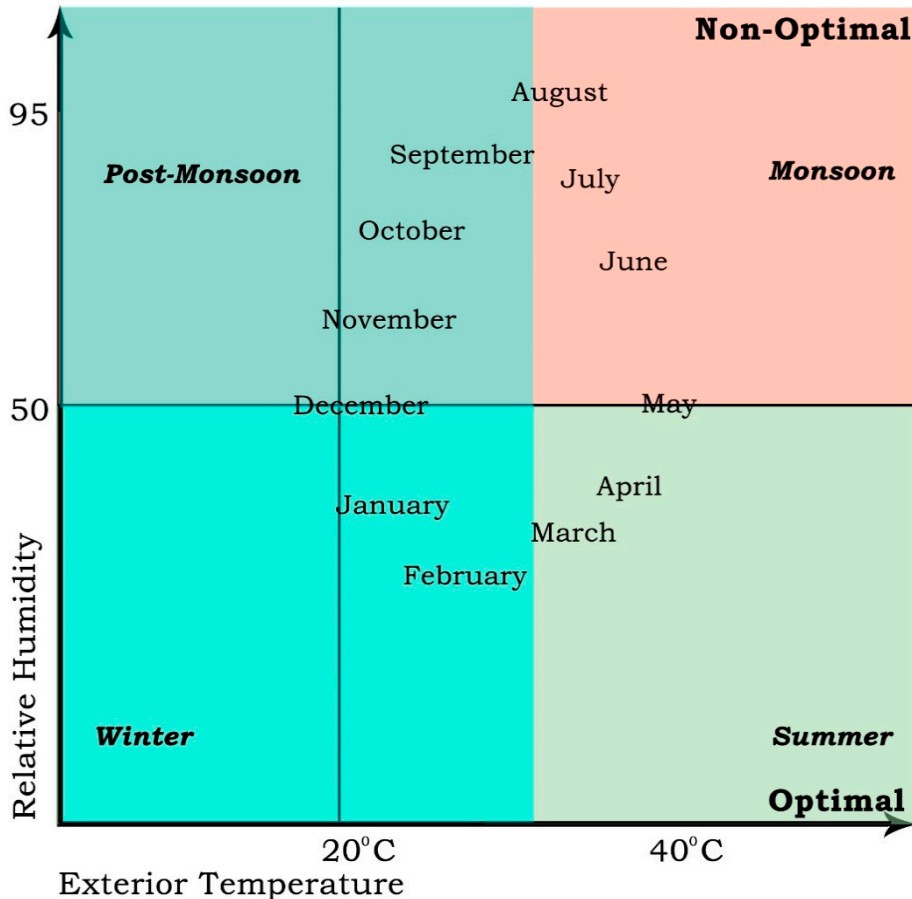

**Figure 5.** This is a graphical representation of the rubric to evaluate optimal and non-optimal conditions for thermal surveys in Hyderabad.

Through the rubric, we understand that the monsoon, primarily, and the months leading to and after the monsoon, from the end of May to early December, are not the most ideal conditions for thermal imaging in Hyderabad. Whereas, periods from the end of December to mid-May are more ideal, owing to the lower relative humidity levels. While this rubric aids with the decision making for thermal imaging planning, sometimes it is not possible to plan surveys in the most optimal conditions alone. The aim of the rubric is to caution the thermographer about the possibility of bad images during non-optimal conditions and to draw more attention to potential misinterpretation of thermal images taken during that period.

## 4. Conclusions and Discussion

Tropical climates are challenging conditions for thermal imaging. The methodologies followed for thermal imaging in colder countries might not apply to tropical conditions, due to different relative humidity and average temperature levels. Through the surveys conducted at the former British Residency in Hyderabad, we were able to understand the effects of these differences. We noticed that an increase in external temperature caused an overall heating of the building, leading to increased evaporation at the surfaces happens in tropical climates, which is different from the solar insolation and induced heating inside buildings, changing the internal and external temperatures of target structures in countries with colder climes.

Another aspect to consider, which is not discussed in detail here, is the change of the rate of evaporative flux that is caused due to the porosity of the material. At the Residency, the bricks and lime mortar are very porous materials, and so the results are more visible. This would not be the same with exposed stone masonry structures. Future work on the rubric could be towards including

more parameters that effect thermal properties of the materials and, consequently, their appearance in the thermal images. When developed sufficiently, this study could be further extended to evaluate the difference in thermal properties of local vernacular materials when used in foreign construction typologies to that of vernacular architecture. This would help understand how vernacular construction materials differ in thermal properties when applied in British design, as opposed to vernacular design and architecture. This is since British or Western designs evolved with the use, maintenance, and repair of their building materials, while in this case, and broadly the subcontinent of India, the use of local Indian materials was widespread. This difference comes to light during the restoration of built heritage and necessitates the establishment of guidelines that are context specific.

**Author Contributions:** Conceptualization, E.R. and M.K.; methodology, E.R.; software, M.K.; validation E.R.; formal analysis, E.R. and M.K.; investigation, M.K.; resources, E.R.; data curation, E.R.; writing—original draft preparation, M.K.; writing—review and editing, E.R.; visualization, M.K.; supervision, E.R.; Both authors have read and agreed to the published version of the manuscript.

**Funding:** This research received no external funding.

**Acknowledgments:** The authors wish to acknowledge the Department of Heritage, Government of Telangana, the Osmania University College for Women, and GN Heritage Matters, the consulting architectural firm for the project of restoration at the Osmania University College for Women. Also, Stephen Caffey and Priya Jain at the College of Architecture, Texas A&M University for their support.

**Conflicts of Interest:** The authors declare no conflict of interest.

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
