# Peer review of "Developing Guidelines for the Use of Passive Thermography on Cultural Heritage in Tropical Climates"

_applsci, doi:10.3390/app10238411_

Round 1

Reviewer 1 Report

The paper aims to provide guidelines to conduct thermographic surveys under non-optimal environmental conditions, such as small indoor-outdoor temperature differences or high relative humidity. This topic, not sufficiently addressed in the literature, might be of particular interest in tropical locations, where the requirements recommended by the thermography standards are hard to meet. However, the academic contribution seems weak in this work, and several aspects of the paper need to improve before consider it for publication:

  • The goal of the paper is outlined in the abstract but not in the body text. Usually, the aim of the study should appear at the end of the Introduction section.

  • The paper lacks some necessary detail for readers to fully understand the case study and its significance. The text should include a description of the Hyderabad climate and provide some additional information about the building, such as its orientation and cross-section (Fig.2). These aspects could help to understand how a possible increase in the outdoor surface temperature due to the absorption of solar radiation could allow for conducting thermography surveys even with little indoor-outdoor air temperature differences.

  • The article contains observations but is not a full study. The data analyzed - two shots in two different days - are too limited to draw sound conclusions or suggest guidelines. The water damages are more evident in thermographies shot under 50% RH than 96%. But what happens for the RH values in-between? Additional experiments could help to enrich the rubric in Fig. 5, which seems arbitrary in its current state.

  • The methodology is unclear and seems inappropriate to accomplish the goal of the study. It seems to me that the authors chose to study two building spots suffering moisture problems to discuss how the environmental conditions affect the defect observation. If this is the case, the analysis should cover a higher number of days.

  • Thermographies in this work use different temperature scales: Celsius and Fahrenheit (Fig. 3 and 4). This kind of representation is misleading. Moreover, the authors should clarify the reflected ambient temperature used for thermography calibration and describe the environmental conditions before the thermography shots (e.g. indoor-outdoor gradient temperature, RH, wind conditions, etc).

  • Lines 195-199: Based on ASTM Standard C1060 we can see that the ideal imaging temperatures result in a delta T between internal and external surface temperatures of 10C. […]. Temperature gradients higher than this result in degraded imaging conditions due to the formation of misrepresentations in the images”.

The ASTM recommends that the indoor-outdoor temperature difference be at least 10°C (18°F) for approximately 4 h before testing. Higher gradients would facilitate the identification of building defects. However, the text above suggests the opposite.

Moreover, this work seems to indicate that it is possible to detect certain building defects with small indoor–outdoor temperature differences (<3°C). This observation deserves further discussion.

  • Please check the English writing significantly. There are a lot of grammatical and punctuation mistakes that make the text hard to understand at some points. For example:

Punctuation mistakes:    

Line 35: “Though, a promising tool,” --- “Though a promising tool, ”

Sentence with no subject:

Line 36: “When onsite, at uncontrolled conditions of air temperature, relative humidity, solar irradiation, etc. affect thermal signatures captured by the thermal camera.”

Incorrect prepositions:

Line 54: “depending of” --- depending on”

Typos:

Line 74: “the building and it’s premises” --- “the building and its premises”

Reviewer 2 Report

The presented article is interesting from the point of view of approaching to infrared thermography as the tool for the diagnosis and monitoring of cultural heritage. The authors note that one of the limitations of using this method is the lack of the desired standardization. Hence the authors suggest the guidelines for the use of passive thermography on cultural heritage in tropical climates.

  • References should be described according to instructions for Authors, mdpi.com/journal/applsci/instructions
  • It appears that authors should take permission for reproduction the foto (figure 1)
  • Temperature units (in table 1 and text) should be corrected; it is ..C, it should be ..oC

Reviewer 3 Report

The manuscript introduces comprehensively some guidelines and suggests an evaluation schema for passive Infrared Thermography (IRT) in historic buildings regarding extreme environmental conditions. The paper serves more as a case study and less as a technical note as the proposed approach is pivoted on tropical climates and a relative thermographic survey, conducted in Hyderabad, India.

The content is interesting and understandable by non-specialists. Although the contribution novelty is not a strength, the article will be of interest to the readers of the Special Issue assigned. The overall evaluation of the paper is quite positive. Nevertheless, some remarks should be referred to it and a revision in this sense would be welcome. Here is a list of points that should be reconsidered:

General Comments

1.The literature review is rather poor/inadequate. Consider to include a short relative section or extend the Introduction.

2.Since an “Implementation” section and sufficient details of the methodology are missing, the experimental results are not supported by the evidence: Due to the technical scope of the Special Issue, the authors should consider mentioning the equipment and software used during the study, the size and resolution of the raster images, their geometric accuracy (if any) and any other information that would determine the documentation process and define a clear workflow.

3.The authors should consider illustrating their workflow with a figure or flow-chart.

Introduction

Line 30: I would advise the authors to supplement the literature review of the section with more recent research and scientific papers, such as:

  1. Lerma, Á. Mas, E. Gil, J. Vercher, and M. E. Torner, “Quantitative Analysis Procedure for Building Materials in Historic Buildings by Applying Infrared Thermography,” Russ J Nondestruct Test, vol. 54, no. 8, pp. 601–609, 2018. Doi: 10.1134/S1061830918080065.
  2. Delegou, G. Mourgi, E. Tsilimantou, C. Ioannidis and A. Moropoulou, “A Multidisciplinary Approach for Historic Buildings Diagnosis: The Case Study of the Kaisariani Monastery,” Heritage. 2019;2(2):1211-1232, 2019. Doi:10.3390/heritage2020079
  3. Adamopoulos, M. Volinia, M. Girotto and F. Rinaudo, “Three-Dimensional Thermal Mapping from IRT Images for Rapid Architectural Heritage NDT,” Buildings 10, 187, 2020. Doi: 10.3390/buildings10100187

Line 33: Consider replacing references [7] and/or [8] with up-to-date research on the topic, such as:

  1. L. Lerma, M. Cabrelles, and C. Portalés, “Multitemporal thermal analysis to detect moisture on a building façade,” Construction and Building Materials, vol. 25, no. 5, pp. 2190–2197, 2011. Doi: 10.1016/j.conbuildmat.2010.10.007.
  2. H. A. Rocha, C. F. Santos, and Y. V. Póvoas, “Evaluation of the infrared thermography technique for capillarity moisture detection in buildings,” Procedia Structural Integrity, vol. 11, pp. 107–113, 2018. Doi: 10.1016/j.prostr.2018.11.015.
  3. Ruiz Valero, V. Flores Sasso, and E. Prieto Vicioso, “In situ assessment of superficial moisture condition in façades of historic building using non-destructive techniques,” Case Studies in Construction Materials, vol. 10, p. e00228, 2019. Doi: 10.1016/j.cscm.2019.e00228.

The introduction presents concisely the general area of interest but it does not establish the originality of the research nor the state of the current knowledge. A reference to the argument, the aim and the contributions of the work presented in this section (for instance in a paragraph after line 66) is be strongly suggested.

Materials

Line 92: Consider to rephrase the sentence.

Line 136: Please clarify the conditions that do not comply with the standards as it is a key point to understand the proposed methodology and the metrics provided in Table 1.

Results

Line 165: Consider to rephrase the sentence.

Line 168: I do not think that this statement is a large inference; it was expected considering the conditions.

Lines 174 - 183: I think that the content of this paragraph is not consistent with the “Results” sections and it may confuse the readers; consider to include a short description of the impact of relative humidity in the “Introduction” section with relative and up-to-date references.

Line 174: It is not well documented unless relative scientific work is provided. Consider to supplement the references of line 178 with up-to-date scientific endeavors.  

Line 196: Consider to rephrase the sentence (verb is missing).

Line 200: Consider to rephrase the sentence.
